# Influence of Hyperproteinemia on Insect Innate Immune Function of the Circulatory System in *Bombyx mori*

**DOI:** 10.3390/biology10020112

**Published:** 2021-02-03

**Authors:** Yong-Feng Wang, Guang Wang, Jiang-Lan Li, Ya-Xin Qu, Xin-Yin Liang, Xue-Dong Chen, Yang-Hu Sima, Shi-Qing Xu

**Affiliations:** 1School of Biology and Basic Medical Sciences, Medical College, Soochow University, Suzhou 215123, China; yfwang1993@suda.edu.cn (Y.-F.W.); 20194021007@stu.suda.edu.cn (G.W.); 20194221011@stu.suda.edu.cn (J.-L.L.); 20174221004@stu.suda.edu.cn (Y.-X.Q.); 20184221025@stu.suda.edu.cn (X.-Y.L.); chenxd@suda.edu.cn (X.-D.C.); simyh@suda.edu.cn (Y.-H.S.); 2Institute of Agricultural Biotechnology & Ecology (IABE), Soochow University, Suzhou 215123, China

**Keywords:** *Bombyx mori*, innate immunity, melanization, NF-κB signaling, plasma protein concentration (PPC)

## Abstract

**Simple Summary:**

Hyperproteinemia, a condition of elevated protein levels in the blood, is associated with a diverse range of human and animal diseases. However, there is no reliable hyperproteinemia disease models or modeling methods in mammal or other organisms, and the effect of hyperproteinemia on immunity is still unknown. Our work succeeded in constructing an animal model of hyperproteinemia with no primary disease effects and a controllable plasma protein concentration (PPC) in an invertebrate model organism, *Bombyx mori*. Our work confirmed that high PPC enhances hemolymph phagocytosis via a rapid increase in granulocytes and inhibited hemolymph melanization due to inhibition of the prophenoloxidase (PPO) signaling pathway, and also upregulated the gene expression of antimicrobial peptides via activating the Toll and Imd pathways in NF-κB signaling, and showed an inconsistent antibacterial activity for Gram-positive and Gram-negative bacteria. Our results show that high PPC had multiple significant effects on the innate immune function of the silkworm circulatory system and is expected to be improved by endocrine hormones. Our work explores the pathogenesis of hyperproteinemia in an invertebrate model, and expands the scope for silkworm biomedical applications, even use for a potential drug development platform.

**Abstract:**

Metabolic disorders of the circulatory system of animals (e.g., hyperglycemia and hyperlipidemia) can significantly affect immune function; however, since there is currently no reliable animal model for hyperproteinemia, its effects on immunity remain unclear. In this study, we established an animal model for hyperproteinemia in an invertebrate silkworm model, with a controllable plasma protein concentration (PPC) and no primary disease effects. We evaluated the influence of hyperproteinemia on innate immunity. The results showed that high PPC enhanced hemolymph phagocytosis via inducing a rapid increase in granulocytes. Moreover, while oenocytoids increased, the plasmacytes quickly dwindled. High PPC inhibited hemolymph melanization due to decreased phenoloxidase (PO) activity in the hemolymph via inhibiting the expression of the prophenoloxidase-encoding genes, *PPO1* and *PPO2*. High PPC upregulated the gene expression of antimicrobial peptides via differential activation of the Toll and Imd signaling pathways associated with NF-κB signaling, followed by an induction of inconsistent antibacterial activity towards Gram-positive and Gram-negative bacteria in an animal model of high PPC. Therefore, high PPC has multiple significant effects on the innate immune function of the silkworm circulatory system.

## 1. Introduction

Hyperproteinemia is a metabolic disease, which has recently been reported to be associated with an abnormally elevated plasma protein concentration (PPC) as a clinical symptom [1,2,3,4,5]. This hyperproteinemia is commonly exhibited in many important clinical diseases in humans, as well as other mammals (e.g., multiple myeloma [4,6], nephropathy [7,8,9,10], hepatopathy [11,12], and pathogen infections [2,13]). Hyperproteinemia also appears in metabolic acidosis and even clinical plasma therapy or intravenous immunoglobulin therapy [9,14,15,16]. A clinical study from the Benghazi Medical Center showed that out of 99 diagnosed cases of multiple myeloma, 30% developed hyperproteinemia (>9 gm/dL) [4]. A total of 51 dogs with canine leishmaniosis (*CanL*) caused by *Leishmania infantum* were studied at the School of Veterinary Medicine of the Autonomous University of Barcelona, and clinical hyperproteinemia was found to be as high as 52.9% [2]. 

Metabolic disorders of the circulatory system are typically characterized by the presence of abnormally increased or decreased metabolites in the circulating blood, which can cause many clinical diseases (e.g., hyperglycemia [17], hyperlipidemia [18], hypercholesterolemia [19,20], and hyperuricemia [21]). It is particularly interesting that patients with hyperproteinemia generally display metabolic changes (e.g., abnormal blood glucose and blood lipid levels) [4,5], and there is a significant change in the proportion of circulating blood cells [8,22]. These findings imply that hyperproteinemia has complex effects on the circulatory system.

Previous studies have shown that a metabolic balance disorder of the circulatory system affects the normal function of blood cells and is closely related to changes in blood immunity [23,24,25]. Hyperglycemia and metabolic disorders may directly promote pathological changes of normal cell functions through modifications of β-N-acetylglucosamine (O-GlcNAc) [26]. In the hyperglycemia microenvironment, high glucose protects pancreatic cancer cells from natural killer cell (NK)-mediated killing by inhibiting the expression of MHC class I chain-related protein A/B (MICA/B), and promotes the immune escape of cancer cells [27].

In patients with hyperlipidemia, Toll-like receptor (TLR) 2 appears to activate the innate immune signaling cascade in platelets by binding to endogenous lipid peroxidation ligands [28]. Hyperlipidemia can induce an increase in T helper cell 17 (Th17) secreting immune proteins in the spleen, which results in an enhanced immune response to modified lipids and aortic inflammation [29]. Hypercholesterolemia can induce T cell expansion, which plays a key role in the adaptive immune response in humanized immunized mice, and upregulates the expression of transforming growth factor beta 1 (TGFb1) in the liver [29,30]. Hypercholesterolemia enhances TLR signaling, activates inflammasomes, and promotes inflammation [31].

Invertebrate model animals, such as *Bombyx mori* and *Drosophila*, have sensitive innate immune systems, including humoral (hemolymph) and cellular immunity, but no adaptive immune system [32,33,34,35]. When the silkworm can resist pathogen invasion, the prophenoloxidase cascade activated in oenocytoids (Oen) by pathogens and other stimuli converts prophenoloxidase (PPO) into active phenoloxidase (PO). PO further hydroxylates monophenolase (MPO) and oxidizes BIS-phenoloxidase, producing a large amount of quinone intermediates to polymerize and form melanin. These melanins then cooperate with cytotoxic quinone intermediates for deposition around invading pathogens, followed by pathogen isolation and killing [36,37,38]. At the same time, granulocytes (Gra) will accumulate in the hemolymph and engulf pathogens. Furthermore, the Toll and Imd pathways in insects and other animal cells are stimulated by microorganisms [32,39,40,41]. Moreover, signaling through these pathways further activate the NF-κB-like transcription factors, Dif/Dorsal, and Relish. These transcription factors are then transferred into the nucleus to induce antibacterial AMPs and the transcription of other related genes. Using *Drosophila* as a model, NF-κB signaling regulation of innate immunity through the Toll and Imd pathways has been widely studied and confirmed [42,43]. Among them, Gram-positive bacteria primarily activate the Toll pathway, whereas the Imd pathway mainly responds to immune stimulation with Gram-negative bacteria [35,44]. There are also bacterial infections that simultaneously activate Toll and Imd, which induce different reactions.

Currently, the effect of hyperproteinemia on immunity remains unclear due to that it is difficult to distinguish hyperproteinemia from primary disease or infection, and there is no reliable animal model in mammals or fruit fly [45,46,47,48]. In addition, hyperproteinemia is often complicated by abnormal blood sugar and blood lipid levels in these patients. In our previous research, we created an animal model of hyperproteinemia (AM) with no primary disease using an invertebrate model animal [45,46], *Bombyx mori*, in which hyperproteinemia could be generated by controlling plasma protein concentration (PPC) levels. This model revealed that high PPC has complex effects on metabolic tissue fat body remodeling and gonad development [45,46]. This suggests that in the open blood circulation system of insects, high PPC may have a direct impact on innate immunity. Hence, we constructed an animal model for hyperproteinemia in silkworm and investigated the effect of hyperproteinemia on innate immune function of the circulatory system in this model.

## 2. Materials and Methods

### 2.1. Animal Model

The Dazao strain of *Bombyx mori* was used as the animal model in the current study. The experimental process to create a hyperproteinemia animal model (AM) was performed as previously described in [45,46]. The larvae were reared on fresh mulberry (*Morus*) leaves at 25 °C with a 12 h light/12 h dark photoperiod until they grew into mature larvae, namely the wandering stage, at which point they were ready to enter the subsequent spinning stage and complete their pupa metamorphosis development. The fibroin in silk glands (SG) (accounts for 30% of the body weight) was then prevented from being released outside the larva by covering the spinneret with a low-melting-point paraffin wax. The fibroin in the glandular cavity is also released into the circulatory system in the subsequent metamorphosis during the physiological degradation of the SG, and then we remodeled a high PPC silkworm (AM). Whereas silkworms in control group (CK) remained untreated. (AM+20E), the larvae of AM were rescued by injecting 4 μg of 20E (10 μL per larva, 0.4 μg/μL) (Sigma-Aldrich, St. Louis, MO, USA) at 24 h after modeling. Then, the larvae were kept under natural conditions as the CK group until they grew into adult moths. 

The hemolymph and fat bodies were collected from the silkworms at the indicated time points after modeling (48, 96, and 192 h), when PPC is in the rising stage, peak stage, and late stage of the platform, respectively. Each tissue (or hemolymph) sample was collected from three female individuals, and each sample was measured three times.

### 2.2. Fat Body Collection and Transcriptome Sequencing

At 48 and 96 h after modeling, fat bodies were removed and collected from each larva in 0.7% NaCl. Total RNA was extracted from fat bodies using the RNAiso Plus Kit (TaKaRa, Dalian, China), chloroform, isopropyl alcohol, 75% ethanol, and DEPC H_2_O, following the manufacturer’s protocol. Transcript library construction and sequencing were performed using Illumina Novaseq 6000 platform by Majorbio (Shanghai, China). Each fat body sample was collected from three female individuals, and each sample was measured three times. The samples were named as CK48h_1/2/3, AM48h_1/2/3, CK96h_1/2/3, AM96h_1/2/3, respectively.

### 2.3. Transcriptome Assembly, Annotation, and Differential Analysis

After sequencing, the original sequencing data were quality controlled to obtain high-quality control data (clean data) to ensure the accuracy of subsequent analysis results. The genome and gene model annotation referenced the silkworm database (http://metazoa.ensembl.org/Bombyx_mori/Info/Index). The expression levels of the gene and new transcripts were assessed by Transcripts Per Million reads (TPM). These differentially expressed genes were subjected to Kyoto Encyclopedia of Genes and Genomes (KEGG) (http://www.genome.jp/kegg/) and Gene Ontology (GO) (http://www.geneontology.org) databases for functional annotation and enrichment analysis. KEGG pathways and GO terms were defined as significantly enriched with *p*-adjust < 0.05. 

### 2.4. Real-Time Polymerase Chain Reaction Analysis

The level of gene transcripts in the hemocytes and fat bodies was analyzed via quantitative real-time polymerase chain reaction (qRT-PCR). Rp49 was used as a reference gene. qRT-PCR was performed in a 20 μL reaction solution using an ABI StepOnePlus™ real-time PCR system (Ambion, Foster City, CA, USA). The total RNA was extracted by traditional methods for qRT-PCR, and qRT-PCR was performed as described previously [46] using the primers listed in Appendix A. 

Hemocyte and fat bodies were removed from the silkworms at the indicated time after modeling (48, 96, and 192 h). Each tissue (or hemolymph) sample was collected from three female individuals, and each sample was measured three times.

### 2.5. Determination of Phenoloxidase (PO) Activity

Hemolymph was collected from the silkworms at the indicated times after modeling, and hemolymph samples were centrifuged under 4 °C (12,000 ×*g* for 5 min). In a 96-well plate, 280 μL PBS was added to each well and preheated at 28 °C for 30 min. Next, 10 μL hemolymph and 10 μL 0.01 mM L-Dopa was added in sequence. An OD value under A490 nm was continuously measured with a microplate reader (20 min period, time interval 2 min, reaction temperature 28 °C). The calculation process was performed as described previously [49]. The PO activity was measured in three female individuals, and each individual was measured three times.

### 2.6. Melanization Measurement

At 48 h after modeling, 50 μL of hemolymph from female silkworms was collected and the changes in hemolymph melanization were observed until one group was completely black at room temperature (25 °C). The observations were repeated in three silkworms (*n* = 3).

### 2.7. Hemocyte Counting

A 5 μL supersaturated solution of phenylthiourea was added to each 200 μL hemolymph sample. Next, 10 μL of the sample was collected and added to the blood count plate. The type and number of hemocytes was counted and observed under a fluorescence microscope (Olympus BX51, Tokyo, Japan) (*n* = 3 samples).

### 2.8. Antibacterial Activity

A total of 20 μL of bacteria, *Escherichia coli* (*E. coli*) and *Staphylococcus aureus* (*S. aureus*), was added to 4 mL of LB liquid culture, respectively. The bacteria were shaken cultured to mid-log phase, then adjusted to OD570 = 0.2 with the culture medium, as the bacterial solution used for measurement. The hemolymph was incubated in a metal bath at 100 °C for 5 min, and then centrifuged under 4 °C at 4000× *g* at 4 °C for 5 min to collect the supernatant plasma. In a 96-well plate, 80 μL bacterial solution and 20 µL plasma was added to each well and shaken cultured (220 r/min) at 37 °C. An OD value under A570 nm as a proliferation parameter of the bacteria was continuously measured using a microplate reader (time interval of 60 min). The measurements were repeated for six wells (*n* = 6), and Kanamycin was added as a positive control. The experimental process was performed as described in [50].

### 2.9. Data Analysis

Statistical analysis of data was performed using GraphPad Prism 8 software and the significance was analyzed using the Holm–Sidak method for multiple t-tests at one per row. Unless otherwise stated, all data were presented as the mean ± SEM (*n* = 3). * *p* < 0.05; ** *p* < 0.01; *** *p* < 0.001; n.s., no significant difference between the two groups. 

## 3. Results

### 3.1. High PPC Induces Hemocyte Phagocytosis

There are five types of hemocytes in the silkworm hemolymph, including prohemocyte (Pro), plasmacyte (Pla), granulocyte (Gra), spherulocyte (Sph), and oenocytoids (Oen) (Figure 1A). Among these hemocytes, the immune function of Gra is phagocytosis, whereas Pla and Oen produce phenoloxidase to encapsulate foreign matter.

The number of Gra and Oen and the ratio of both among the total hemocytes increased rapidly in the AM group at 48 h after modeling. The number of Pla and the ratio in the total hemocytes were seriously reduced (Figure 1B–F). At 48, 96, and 192 h after modeling, the proportion of Gra increased from 26.9%, 28.6%, and 43.5% in the CK group to 56.6%, 54.5%, and 86.3%, respectively, which nearly doubled (Appendix A). The proportion of Oen increased from 0.9% and 2.4% in the CK group to 9.1% and 12.9% at 48 and 96 h, respectively, but dropped from 8.7% in the CK group to 2.4% at 192 h. The most significant reduction was observed for Pla, as its proportion was reduced by 96.4% and 69.1% compared with the CK group at 48 and 96 h, respectively. By 192 h, there was almost no Pla in the AM group, and the proportion of Pla in the CK group was as high as 25.2% (Appendix A). In the hemolymph at 48 h after modeling, it was clearly observed that the increased Gra in the AM group enhanced the envelopment and phagocytosis to the remodeling tissue fat body (FB) (Figure 1C,E). Phagocytosis is a receptor-mediated, actin-dependent endocytosis [51,52]. We investigated the mRNA level of phagocytosis-related marker gene *Actin A1*. The relative mRNA level of *Actin A1* was significantly upregulated in hemocytes compared with CK after modeling (Figure 1G). These results suggest that high PPC may affect the cellular immunity of *Bombyx mori*.

### 3.2. High PPC Affects the Expression of Innate Immune-Related Genes

To evaluate the impact of high PPC on the silkworm innate immune system, transcriptomic sequencing was performed using FB, an important tissue related to anabolism and immunity. The GO analysis showed that at 96 h after modeling, the level of gene transcription of biological process, cellular components, molecular function, and other processes exhibited significant changes, including 21 immune stress-related genes and 19 immune system process genes directly related to immune function (Figure 2A).

The transcriptome immunoassays showed that at 96 h after modeling, antimicrobial peptides (AMP) genes (e.g., *CECA*, *LEB1/2*, *BGIBMGA013108*, and *BGIBMGA014285*) were significantly upregulated compared with the CK group (Figure 2B). When comparing 96 h to 48 h after modeling, the number of immune-related genes that changed significantly increased from 10 to 21, 18 of which were newly emerged genes (Figure 2C). The heatmap shows that 21 immune-related genes exhibited significant changes, except for the downregulation of *BGIBMGA006716* and *BGIBMGA007400*, the remaining 19 genes are all significantly upregulated (Figure 2D).

Further KEGG analysis revealed that high PPC significantly induced the expression of AMP genes in innate immune response. Moreover, genes in the Toll and Imd signaling pathway were involved in the regulation of the expression of AMPs in NF-κB signaling, which exhibited significant changes in the level of transcription (Appendix A). In addition, high PPC induced changes in silkworm innate immunity.

The level of AMP mRNA in the FB was measured (Figure 3), and it was found that at 48, 96, and 192 h after modeling, the level of *Moricin*, *Defensin*, *Lebocin*, and *CecropinA* mRNA expression was significantly upregulated in the AM group, in which *Defensin* was upregulated to the greatest extent (Figure 3C–F). At the same time, the level of *Attacin* and *Gloverin* mRNA expression was significantly downregulated compared with CK at 96 h after modeling; however, a significant upregulation or no difference was observed between 48 and 96 h (Figure 3A,B). The differences in AMP transcription in the AM group suggest that it may be related to different upstream signaling pathways that regulate AMP expression. It is important to note that in both the CK group and AM group, the level of AMP mRNA in the FB first increased then decreased during the 48–192 h period (Figure 3). This finding is highly consistent with the PPC changes in the AM group reported in our previous modeling experiments. This implies that the level of AMP mRNA in the FB is affected by the level of PPC.

### 3.3. High PPC Affects Hemolymph Antibacterial Activity

To further examine the impact of high PPC on the innate immune system, we further focused on the silkworm hemolymph, which exerts the most extensive innate immunity, and investigated the humoral immune response to high PPC.

The antibacterial activity testing results using growth rate as an indicator (Figure 4A and Appendix A) showed that the plasma of the AM group had a significant effect on the antibacterial activity of the Gram-negative bacteria, *Escherichia coli* (*E. coli*). Although there was no statistical difference between the AM group and the CK group at the early time point of 48 h after modeling, the growth rate of *E. coli* in the AM group was significantly lower than that of the CK group at 96 and 192 h after cocultivation for more than 2 h (Figure 4A and Appendix A).

However, the Gram-positive bacteria, *Staphylococcus aureus* (*S. aureus*), exhibited different results. At 48 h after modeling, the growth rate of the AM group was slower than that of the CK group during 1–3 h of coculture. However, no difference was observed between the 4–6 h coculture and the CK group (Appendix A). In the 96 and 192 h investigations, the growth rate of the AM group did not differ from that of the CK group within 3 h of cocultivation. The continued culture was significantly faster than that of the CK group (Figure 4A and Appendix A). This finding showed that high PPC induced an increase in the antibacterial activity of silkworm hemolymph against *E. coli*, but inhibited the antibacterial activity against *S. aureus*.

The level of AMP mRNA, the main humoral immune substance in silkworm hemocytes, was further determined. It was found that at 48, 96, and 192 h after modeling, the level of *Attacin, Gloverin, Moricin, Defensin, Lebocin*, and *CecropinA* mRNA was significantly upregulated, except for the downregulation of *Gloverin* gene transcription at 96 h. Moreover, the *Defensin* and *CecropinA* genes were upregulated hundreds to thousands of times compared with that of the CK group (Figure 4B). 

It is important to note that the degree of induction of AMPs expression in hemocytes of the AM group was much higher than that in the FB as a whole (Figure 3 and Figure 4). This further confirmed that high PPC may have a major impact on the innate immunity of hemocytes by inducing the expression of AMPs. In the results presented in Figure 3 and Figure 4, the differences in the transcription level of the AMP members in the AM group suggest that high PPC may be related to differences in the effects of different upstream signaling pathways, which regulate the expression of AMP members.

### 3.4. AMP Expression is Regulated by NF-κB Signaling

We further investigated the changes in transcription levels of key NF-κB signaling molecules involved in regulating AMP expression in the hemocytes (Figure 5 and Appendix A). The key member gene mRNA levels of the Toll pathway showed that *spaetzle-1 (Spz-1)*, *Toll-4*, *Myd88,* and *Dorsal* in the hemocytes exhibited significant changes after modeling. During the three time points of investigation, with the exception of a continuous increase in the level of *Dorsal* mRNA, the level of transcription of the other three genes in the CK group all showed changes that first increased and subsequently decreased. Unlike the CK group, the mRNA levels of these four genes in the AM group were rapidly downregulated from 48 to 192 h after modeling (Figure 5). From the perspective of gene transcription, the mRNA levels of these four genes in the AM group were significantly upregulated dozens of times more than CK, early at 48 h after modeling. Among these, the Dorsal transcription level of the Toll signal was upregulated by 250-fold. However, in the late time point of 192 h after modeling, the level of *Spz-1*, *Myd88*, and *Dorsal* transcription in the AM group were all downregulated to levels lower than that of CK. Moreover, *Toll-4* was also significantly downregulated to a level close to that of the CK group (Figure 5D).

Next, we investigated the changes in the level of *PGRP-S2*, *Imd, Dredd*, and *Relish* mRNA expression in the Imd pathway in the hemocytes (Appendix A). With the exception of *Dredd*, the transcription levels of the other three genes of the AM group, as well as changes in the trends during the period 48–192 h after modeling was significantly different from that of the CK group. Most of the experimental time points showed that high PPC induced upregulation of *PGRP-S2*, *Imd, Dredd*, and *Relish* (Appendix A). Similar to the changes in the levels of gene transcription in the Toll pathway, *PGRP-S2*, *Dredd*, and *Relish* in the control group revealed a trend of first increasing followed by a decrease at 48, 96, and 192 h after modeling; however, the overall expression level was lower (Appendix A). The significant difference in gene transcription of key members of the Toll and Imd pathways in the AM group suggests that high PPC has a greater impact on the Toll pathway associated with NF-κB signaling in hemocytes compared to the Imd pathway.

### 3.5. High PPC Affects Hemolymph Melanization

The aforementioned results presented in Figure 1 show that the proportion of Pla and Oen that produce phenoloxidase in the hemolymph of the AM group changed significantly. This finding implies that melanization plays an important role in the hemolymph immunity. Further investigation confirmed that the hemolymph of the AM group was naturally exposed to the environment in vitro, and the melanization rate was slower than that of control CK group (Figure 6A). It is speculated that the hemolymph of the AM group had a compensatory effect on the production of phenoloxidase though a significant decrease in Pla and increase in Oen. Next, we investigated the key enzymes involved in melanin production in the hemolymph. The phenoloxidase (PO) activity of the AM group was significantly lower than that of the CK group at 48–192 h after modeling. At 192 h, the PO activity reached almost trace levels (Figure 6B). The PO activity of the CK group decreased for a short time at 96 h, and was completely restored at 192 h after modeling. 

We understand that this observation is related to the increase in hemolymph immune consumption caused by the FB remodeling process at approximately 96 h. The transcription levels of the two *prophenoloxidase* (*PPO*) genes in the hemocytes of PO protein synthesis tissue in hemolymph were investigated. Although the level of *PPO1* and *PPO2* mRNA expression was also downregulated with physiological remodeling of the FB in the CK group, the AM group at 48–192 h after modeling was always significantly lower than the CK group (Figure 6C,D). It shows that high PPC inhibits the activity of PO and weakens the melanization of hemolymph by downregulating the mRNA expression of *PPO1* and *PPO2*.

Our previous research results showed that hyperproteinemia may weaken the 20E signaling pathway during silkworm metamorphic development [32]. To explore the relationship between the effect of high PPC on the silkworm hemolymph innate immune response and the endocrine system, we performed a rescue experiment of endocrine hormone 20E on the melanization induced by hyperproteinemia. At 24 h after the injection of 20E (48 h after modeling), the level of *PPO1* mRNA expression and PO activity were restored to the level of the CK group (Figure 6E,F). The rate of hemolymph melanization in the 20E rescue group was significantly faster than that of the AM group, and even faster than that of the CK group (Figure 6A). This finding shows that the melanin effect induced by hyperproteinemia is weakened and can be effectively rescued by an exogenous supplementation with 20E.

To explain the weakening of hemolymph melanization in the AM group, we further investigated the content of serine, a representative metabolite of the serine protease cascade system that affects the initiation of insect innate immune responses, and the changes in the transcriptional activity of serine protease inhibitor (Serpins, Spns) genes and hemolymph protease (HPs) gene in hemocytes (Appendix A). *Spns* and *Hps* have an important effect on PO activity. In the 192 h experimental time period after modeling, the level of *Spn5* gene transcription in the AM silkworm blood cells was higher than the control except at 96 h, and the overall trend was significantly downregulated (Appendix A). The level of *Spn5* gene transcription was downregulated at 48 h and rebounded with an upregulation at 96 h, which can be explained by a significant increase in the hemolymph serine content. The feedback regulation of recovery at 96 h, but the downregulation of the level of *Spn5* gene transcription at 192 h may be related to the complex effect of high PPC (Appendix A). Investigation of the level of *prophenoloxidase activating enzyme* (*PPAE*) gene transcription in relation to the regulation of *Spn27A* transcription revealed that the decrease in hemolymph melanization ability is consistent with the change in the level of *PPAE* gene transcription induced by a high PPC environment (Appendix A). The transcription level of another *hemolymph protein* (*HP*) gene, which can affect the activity of PPO, was significantly higher than the control at 48 h after modeling; however, it was significantly lower than that of the control at 96 and 192 h (Appendix A). This finding implies that there is a complex regulatory mechanism involved in the weakening of hemolymph melanization in the AM group.

## 4. Discussion

Abnormal metabolism of the circulatory system has a significant adverse effect on the immune system of mammals [53,54]; however, there remains a lack of reports on the effect of high PPC on the immune system in the circulatory system. It has been reported that hyperglycemia can induce the expression of Tolls in blood cells, inhibit neutrophil migration and apoptosis, as well as reduce its phagocytosis among other functions [53].

According to previously established methods [45], this study constructed a hyperproteinemia silkworm animal model (AM) and found that Gra and Oen in the circulating hemolymph of the AM increased significantly, whereas Pla was significantly reduced. High PPC also significantly affected the gene expression of the innate immune system, weakened melanization. In addition, the antibacterial activity of hemolymph against Gram-negative bacteria and positive bacteria has undergone inconsistent changes. These findings show that hyperproteinemia significantly affects the silkworm innate immune function and demonstrates the complexity of the pathway.

### 4.1. High PPC Regulates Innate Immunity via the NF-κB Signaling Pathway

In the bodily fluids of insects (e.g., silkworms), AMPs, antiviral factors, and agglutinin play an important role in immunity [55,56], and AMPs are the key substances involved in humoral immunity. AMPs found in insects primarily include Attacin, Cecropin, Defensin, Gloverin, Moricin, and Lebocin [57]. AMPs are mainly synthesized in the FB and hemocytes, followed by secretion into the hemolymph [58].

Studies have shown that the Toll and Imd pathways in insects and other animal cells are stimulated by microorganisms [32,39,40,41]. Peptidoglycan (PGN) recognition proteins (PGRPs) specifically activate the Toll pathway in response to lysine PGN (Lys-PGN) in most Gram-positive bacteria, and IMD pathway in response to diaminopimelic acid (DAP) PGN in Gram-positive bacteria and Gram-negative bacteria [59]. Reports of the relevant recognition mechanisms showed that peptidoglycan recognition protein S2 (PGRP-S2) is the main pattern recognition receptor of the Imd pathway. Immune stimulation is recognized by *PGRP-S2*, which activates *Imd* and *Dredd* in the membrane and in turn, regulates Relish transcription [60]. In the silkworm, the Fas-related factor, FAF, negatively regulates the Imd pathway and regulates antibacterial immunity by binding and degrading Relish [61].

Most AMPs in silkworm are activated by microbial stimuli. Cecropins are the most sensitive, exhibiting high levels of expression and antibacterial activity [62,63]. Among the antimicrobial peptides (AMPs) of silkworm, Attacin, Gloverin, and Lebocin only showed strong antibacterial activity against Gram-negative bacteria, while Cecropin, Moricin, and Defensin had inhibitory effects on both Gram-negative bacteria and Gram-positive bacteria [55,57,64,65]. Therefore, AMPs induced by high PPC significantly inhibited the growth of Gram-negative bacteria *Escherichia coli* (*E. coli*), but had little effect on Gram-positive bacteria *Staphylococcus aureus* (*S. aureus*).

The results of this study show that high PPC induced the significant upregulation of the six *AMPs* genes investigated in the fat body and hemocytes. We also found differences in the types of AMPs regulated by the Toll and Imd pathways. Further tests of the hemolymph antibacterial activity confirmed the presence of AM hemolymph that can significantly inhibit the Gram-negative bacteria, *E. coli*; however, this effect was not observed for the Gram-positive bacteria, *S. aureus*. This finding indicates that the difference in hemolymph antibacterial activity induced by high PPC against different pathogens may be related to AMP regulation via the Toll and Imd pathways in the immune response.

### 4.2. High PPC Inhibits Hemolymph Melanization in Silkworm

Humoral melanization is an important method of innate immunity in insects and other animals. In mutant fruit flies with low PO activity, a lower resistance to fungi, Gram-positive, or Gram-negative bacteria, and significantly shorter life span was observed [34,37]. In the high PPC AM silkworm in this paper, the melanization ability of the hemolymph becomes weak, and the activity of the terminal rate-limiting enzyme, PO, is significantly reduced in the melanin synthesis pathway. The level of mRNA transcription of the two encoding genes of the PO proenzyme, *PPO1* and *PPO2*, are also significantly decreased. This finding shows that high PPC weakens melanization of the hemolymph by inhibiting melanin production. This finding is consistent with our previous report that the life span of the high PPC silkworm is significantly shorter [45]. Meanwhile, the results showed that high PPC decreased the number of plasmacytes and reduced the production of phenoloxidase, but the increase of Oen formed a compensation effect for the production of phenoloxidase. Therefore, it is also possible that what high PPC affected was just the number of hemocytes and it did not affect the signaling pathway for the phenoloxidase production in hemocytes.

Serine is a representative metabolite of the serine protease cascade system initiated by insect innate immune response, and the content of serine affects the changes of serine protease (Sp) and Serpins. Studies have shown that the serine protease cascade plays a key role in the initiation of the innate immune response, such as melanization and antimicrobial peptide production, and is strictly and precisely regulated by serine protease inhibitors (Serpins) [66]. The transcription of the two encoding genes of PPO, PPO1 and PPO2, are regulated by Serpins and HPs [34,67,68]. *Serpin-5* (*Spn5*) of the Serpins family inhibits the silkworm blood and body melanization by inhibiting PPO activity [69]. Spn5 and Serpin-9 (Spn9) inhibit cSP4 and cSP6 activity of the clip-domain serine protease (cSP) family to prevent PPO activation, which inhibits melanization in the cotton bollworm [70]. Serpin-27A (Spn27A) regulates the melanization cascade in *Drosophila* by specifically inhibiting PPAE activation of PPO [71]. In this study, in addition to the increase in serine content derived from silk protein in the early stage after modeling, the feedback regulation of Spns activity and the level of gene transcription during the content recovery process shows that the level of *Spn5* and *PPAE* transcription in hemocytes was associated with a trend of downregulated expression. Furthermore, the level of hemolymph protease *HP4* gene transcription was significantly downregulated 48 h after modeling. Together, these results show that high PPC leads to the downregulation of PPO anabolism in the hemolymph, including its feedback inhibition pathway-related *Serpins* and *HPs* and other transcription factors. This effect is a manifestation of reduced sensitivity to a high PPC response.

Previous studies have shown that melanization is closely related to the Toll pathway in NF-κB signaling. In the *Drosophila* PPO deletion mutants, *PPO1^Δ^* and *PPO2^Δ^*, the lack of melanization specifically enhances Toll activation, despite the findings that PPO is not required for Toll and Imd pathway activation [37]. In the silkworm, knocking down the Toll ligand, Spätzle3 (*Spz-3*), reduced the deposition of epidermal melanin, whereas the ectopic expression of *Spz-3* induced the deposition of epidermal melanin [72]. Since NF-κB signaling is highly conserved in both insects and mammals [73,74,75], it is implied that the high PPC in the silkworm model in relation to NF-κB signaling may have a broader significance for melanization. In the present study, the level of *Spz-1* transcription in the Toll pathway in the hemocytes increased during the initial 48 h after modeling, and significantly decreased thereafter. However, our previous research results showed that the blackening of the epidermis of the high PPC silkworm was significantly more severe than the control after 96 h of modeling [45]. This suggests that the weakening effect of high PPC on the melanization in the hemolymph of silkworm may be related to enhanced melanization of the epidermis during the same period.

## 5. Conclusions

High PPC has multiple significant effects on innate immune function via changes in the composition of the hemocytes, inhibition of hemolymph melanization, differential activation of the Toll and Imd pathways in NF-κB signaling (Figure 7).

## Figures and Tables

**Figure 1 biology-10-00112-f001:**
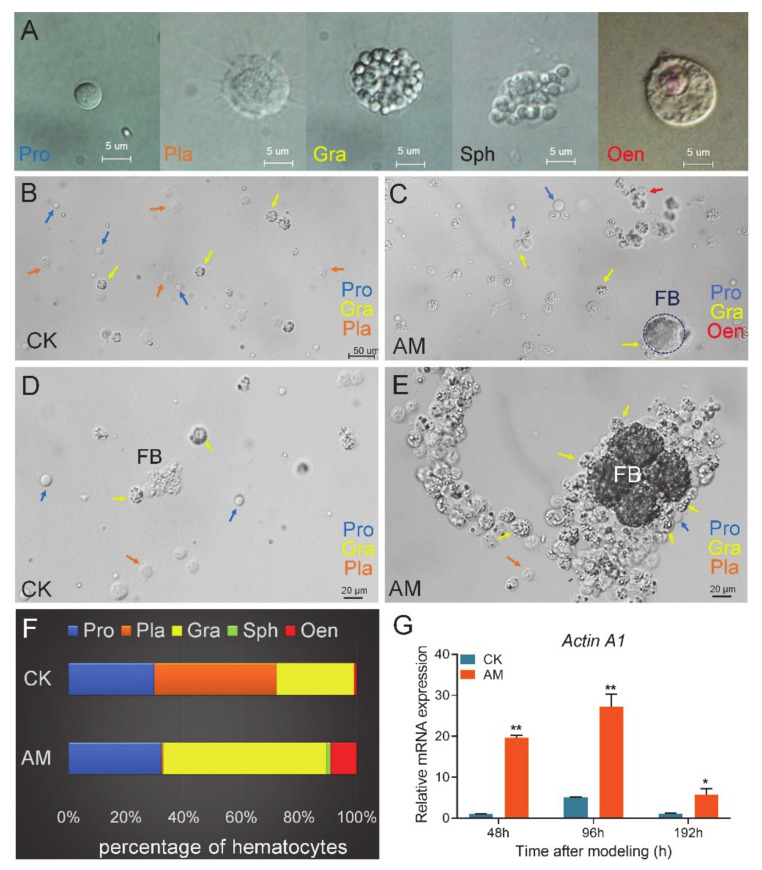
Effect of high plasma protein concentration (PPC) on hemocytes in hemolymph of silkworm. (**A**) Types of hemocytes in hemolymph of the silkworm pupae. Pro, prohemocyte. Pla, plasmacyte. Gra, granulocyte. Sph, spherulocyte. Oen, oenocytoid. (**B**,**C**) The state and classification of hemocytes status in CK group (**B**) and AM group (**C**) at 48 h after modeling. (**D**,**E**) A large number of granulocytes gather around the free fat body in AM group (**E**) compared to the CK group (**D**) at 48 h after modeling. (**F**) The proportion of hemocytes at 48 h after modeling. (**G**) The relative mRNA levels of phagocytosis-related marker gene *Actin A1* in hemocytes from silkworm at 48–192 h after modeling. * *p* < 0.05; ** *p* < 0.01. Mean ± SEM. *n* = 3.

**Figure 2 biology-10-00112-f002:**
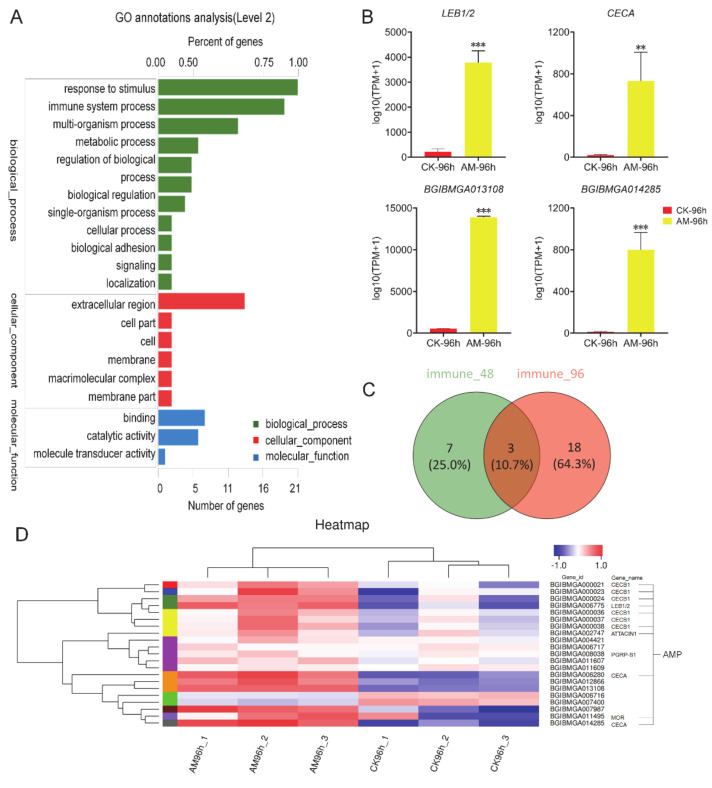
Transcriptome analysis of immune-related genes. (**A**) GO annotations analysis at 96 h after modeling. ‘Number of genes’ and ‘Percentage of gene’ refer to the number of differentially expressed genes involved in biological processes and its proportion in the total number of differentially expressed genes, respectively. (**B**) The expression of immune-related genes changed significantly at 96 h after modeling. (**C**) Venn diagram, the number of immune-related genes that changed significantly at 48 and 96 h after modeling. (**D**) Heatmap, differences in expression of immune-related genes that significantly changed at 96 h after modeling. Most of the genes belong to AMP genes, and others have not been annotated. Each column in (**D**) represents a sample, and each row represents a gene. The color represents the normalized expression value of the gene in each sample. Red represents the higher expression level of the gene in the sample, and blue represents the lower expression level. The number label under the color bar on the upper right represents the specific expression level change trend. On the left is the tree diagram of gene clustering and the module diagram of sub-clusters, and on the right is the name of the gene. ** *p* < 0.01; *** *p* < 0.001. *n* = 3.

**Figure 3 biology-10-00112-f003:**
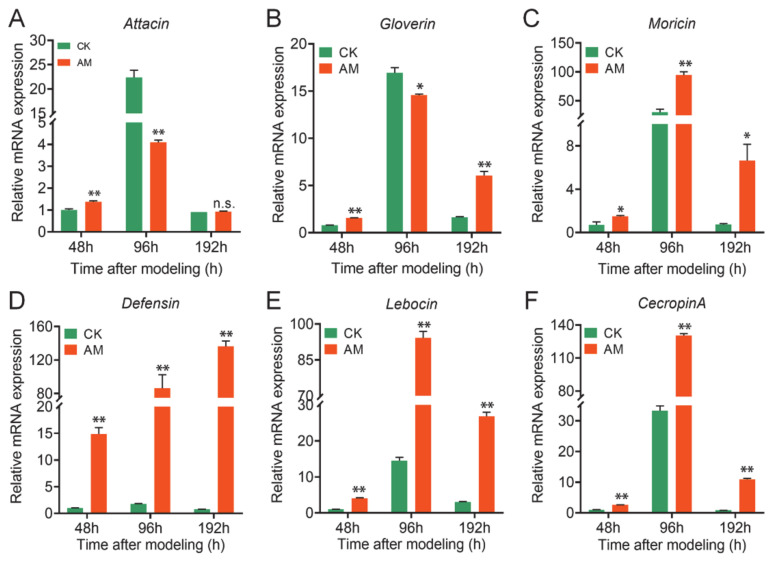
High PPC induced AMPs upregulation in fat body. (**A**–**F**) mRNA levels of AMP genes in fat body from silkworm at 48–192 h after modeling. * *p* < 0.05; ** *p* < 0.01; n.s., no significant difference between the two groups. Mean ± SEM. *n* = 3.

**Figure 4 biology-10-00112-f004:**
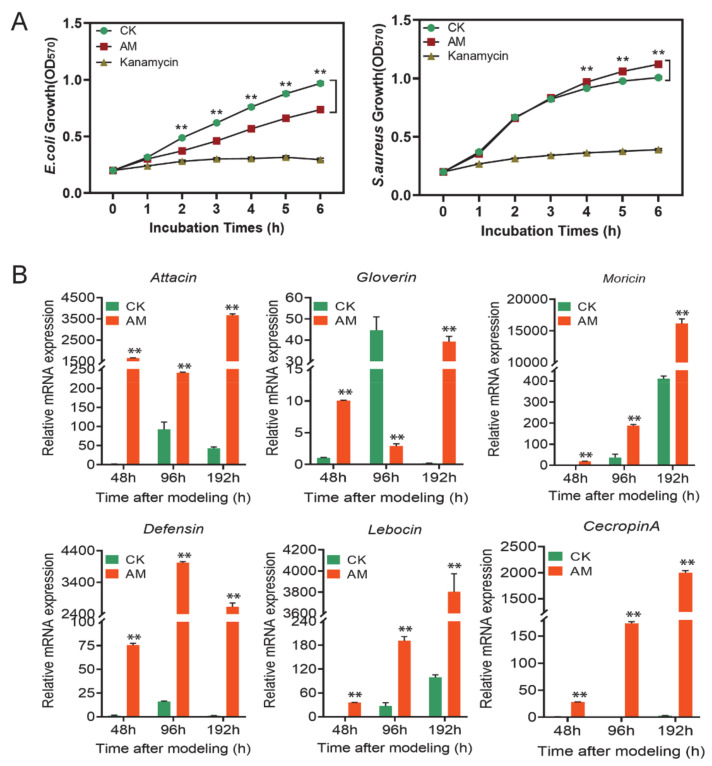
High PPC affected the antibacterial activity in hemolymph. (**A**) Antibacterial activity of high PPC silkworm cell-free hemolymph. The hemolymph of the silkworm was removed for heat treatment at 192 h after modeling, and the supernatant was cocultured with *Escherichia coli* (*E. coli*) and *Staphylococcus aureus* (*S. aureus*), and the growth of *E. coli* (left) and *S. aureus* (right) was investigated. (**B**) The mRNA levels of AMP genes in hemocytes from silkworm at 48–192 h after modeling. ** *p* < 0.01. Mean ± SEM. *n* = 3.

**Figure 5 biology-10-00112-f005:**
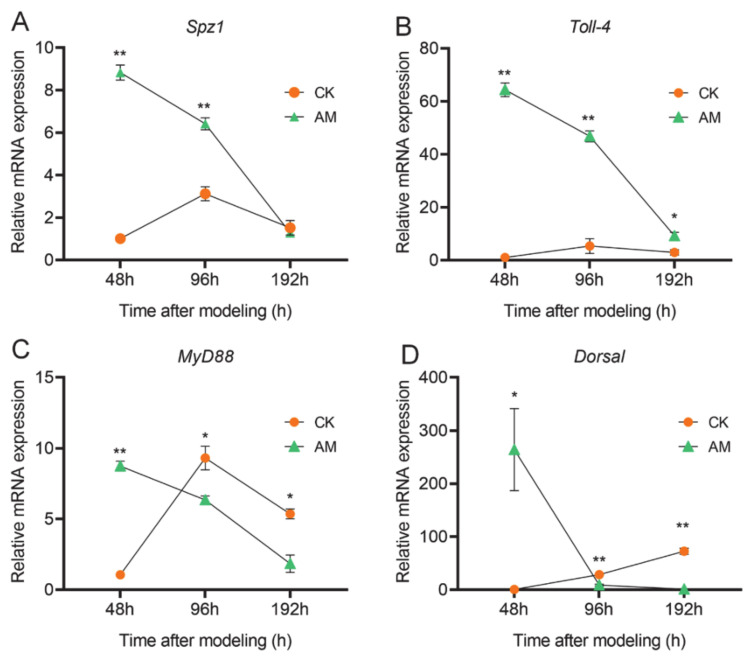
High PPC affected Toll signaling pathway genes expression. The mRNA levels of (**A**) *Spz1* and (**B**) *Toll-4* and (**C**) *Myd88* and (**D**) *Dorsal* genes in hemocytes detected by qRT-PCR. * *p* < 0.05; ** *p* < 0.01. Mean ± SEM. *n* = 3.

**Figure 6 biology-10-00112-f006:**
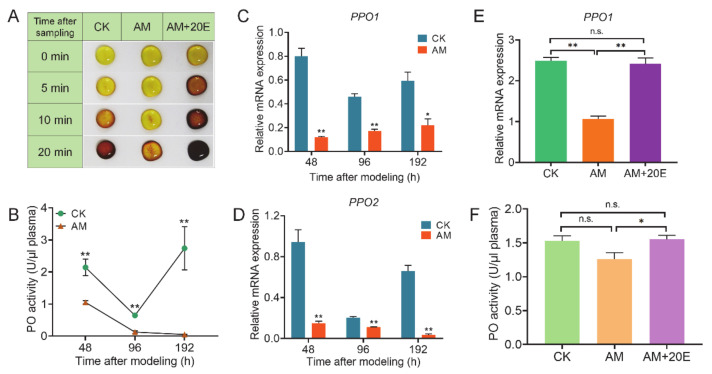
The melanization of hemolymph was inhibited by high PPC in silkworm after modeling and could be rescued by exogenous 20E. CK, the control group; AM, animal model of hyperproteinemia in which hyperproteinemia was induced at the wandering stage; AM+20E, the larvae of the AM were rescued by injecting 4 μg of 20E at 24 h after the induction of hyperproteinemia. (**A**) Melanization speed in vitro of hemolymph. (**B**) Phenoloxidase (PO) activity in hemolymph. (**C**,**D**) The mRNA levels of *PPO1* and *PPO2* in hemocytes. (**E**) The mRNA levels of *PPO1* in hemocytes at 48 h. (**F**) PO activity in hemolymph at 48 h after modeling. * *p* < 0.05; ** *p* < 0.01; n.s., no significant difference between the two groups. Mean ± SEM. *n* = 3.

**Figure 7 biology-10-00112-f007:**
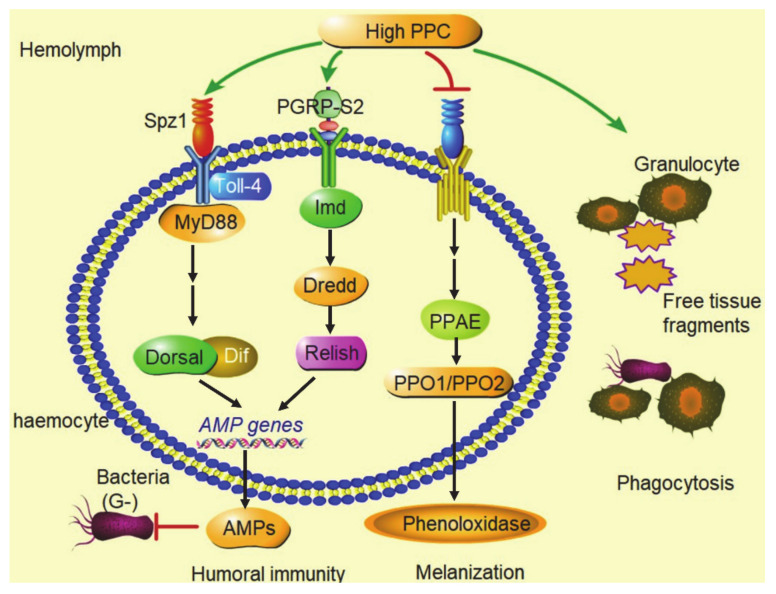
Summary graph. High PPC inhibited phenoloxidase production and melanization of hemolymph in silkworm; high PPC induced granulocytes to phagocytize free tissue fragments in hemolymph, such as dissociated fat body cells; high PPC activated the Toll pathway and Imd pathway in NF-κB signaling, and promoted the large expression of AMPs genes.

## Data Availability

The data presented in this study are available on request from the corresponding author.

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
