# Peer review of "Influence of Hyperproteinemia on Insect Innate Immune Function of the Circulatory System in *Bombyx mori"

_biology, 2021, doi:10.3390/biology10020112_

Round 1

Reviewer 1 Report

To Authors,

Wang et al. insisted on the B. mori hyperproteinemia model effect to innate immunity system, which includes the cellular and humoral immunity system, by evaluating the immune factor in this manuscript. I think this viewpoint is valuable to the study for the elucidation of the system to the hyperproteinemia. However, I think the authors did not show enough evidence for their findings. If the authors insist on the current aim, the authors should add more data to this manuscript because the authors insist that the high concentration of blood proteins affects the innate immune system in B. mori. Also, melanin production is affected by the different kinds of amino acids in the hemolymph (blood). With this data in place, the evidence will be more assertive and more persuasive. Therefore, the current data need more evidence of the immune process. By adding these data, I think the authors can reasonably assert their present purpose.

Major comments;

  1. The authors mentioned the high PPC induces phagocytosis by examining the existence ratio of the five types of hemocytes in this manuscript. Although the authors observed phagocytosis's phenomenon to the fat body cells in the AM group, this phenomenon may be a coincidence. To clarify this process was unique to the AM group, it is necessary to show the frequency of phagocytosis using biological replication and examination for phagocytotic activity in the AM and CK group hemocyte.
  2. The authors measured PO activity in the hemolymph of the AM and CK group. The authors did not examine the activation phase using luteus or E. coli in this assay. The reaction curves start with a log phase follows by a linear phase. Linear regression is used to calculate the changes in absorbance per minute during the linear phase of each reaction (this calculation uses the activate reaction and non-activated reaction data). One unit of phenoloxidase activity is defined as ΔA470 = 0.001/min (Gorman and Kanost IBMB 2007).
  3. Additionally, PO activity is known to have large individual differences. The immune response is different in each mori. Thus, the authors need to measure PO activity, one individual, by one individual. Not mix several individuals in PO activity.
  4. The production of melanin is effect by the different kinds of amino acids. If the authors can show which kind of amino acid is more presented in the AM group's hemolymph, the readers can further understand how the high PPC relates to the innate immune system.
  5. I did not find the method for the transcriptome analysis. If the authors use previous data in the current manuscript, the authors should mention that in the methods section.

Minor comments;

  1. Generally, we can't examine the qRT-PCR experiment using hemolymph (line 105) because we can’t get total RNA from the hemolymph. Did you want to express here as "hemocyte"?
  2. Which is correct upregulate (line 18, 34) , or up-regulate (line 76, 186, 191, 196, 242, 243)? Please unify.
  3. Did you use KEGG analysis for GO annotation in the figure2A? If you employed that, please add information in the methods section.
  4. The authors examined melannization in AM, AM+20E, and CK group. Please clarify the reason for using only female mori and reaction temperature.

Author Response

Thank you very much for your comments concerning our manuscript. Those comments are all valuable and very helpful for revising and improving our manuscript, as well as the important guiding significance to our researches. We have tried our best to improve the manuscript and have made a lot of changes which we hope meet with approval.

Reviewer 2 Report

In the manuscript, Wang et al. Follow up on previous work from some of the same authors (refs 32 and 33) , where an insect model for hyperproteinemia was established and its effects on metabolism and development studied. A major selling point for the model, which is based on the obstruction of the salivary glands by paraffin is the lack of any disease-related symptoms other than the hyperproteinemia itself allowing them to study its effects in isolation. In the present manuscript the authors assess the effects on immunity using a range of immune assays, which altogether provide a broad view of immune dysregulation. Again, the argument that the observed effects are primarily due to the hyperproteinemia is used.

At this stage I have some major concerns about the work, which follows up on these arguments: in the previous work the effects of the SG obstruction on the fat body and on reproductive development are described. In my view, it is not clear whether these should not be regarded as disease symptoms and what is the causal relationship between these symptoms and hyperproteinemia. The same applies for the effects on immunity, some of which could for example be a consequence of the fat body defects or the regulation by ecdysone as previously described by the authors and mentioned in this manuscript. This may lead to asynchronous development and deregulation of aspect of immunity. For example in Drosophila some AMPs are strongly regulated during the transition between larval and pupal life (Flybase). 

Fig. 2: It seems to me that parts B and C describe a reaction against fat body (which is affected in the AM larvae) and so does part C, possible an encapsulation (is that what is meant by envelopment), even followed by melanization in part D. Phagocytosis is not described here, this would have to do be done by observing the uptake of (labeled) particles or bacteria and determining a phagocytic index.

Other major points

The model they use is not sufficiently described one has to read the previous papers to get a full understanding of it.

The transcriptome study in Fig. 2 is not described in the Material and Methods section: was this a transcriptome-wide study or only on targeted genes (in which case the GO enrichment is not informative).

In addition, the language is quite poor and in cases misleading, some examples:

Line12: "as you know" is colloquial English

In the same paragraph “this study” is repeated several times, replace some (for example “our work”), similar inconsistent in line 36

Line 19: Incongruous sounds quite negative perhaps mention there is some specificity

Line 52: the affiliations of the authors is misplaced in this part

Line 60 and 61: what is meant with blood system, better “circulatory system” or “immune system” if one wants to be more specific.

Line 80: “Because” introduces side sentences not a main sentence.

Line 89 what does “we reconstructed” mean? From what I understand the same model was used

Line 97: “AM+20E” should be at least in brackets and as indicated before the model is poorly described.

Line 133: Shaken (not shake)

Line 162 should read Gra

Line 164/165: This sentence sounds strange, should be more specific.

Line 192: I do not think it was the KEGG analysis that revealed the immune function for AMPs rather their known immune function was the basis for the GO enrichment seen before.

Line 245, the absolute expression levels were not measured here rather the degree of induction

Lie 288: blood immune system, what does that mean: humoral immunity? Also previous research has clarified the importance of melanization in insect immunity, for example by the David Schneider group (Stanford).

In fig. 2D the genes should be described more properly saying what class they belong to (only in supplement). In part A and B, it should be mentioned what the values were normalized to (what was set to 1). This applies to other figures too.

Fig. 6 I would suggest to arrange the figure part in the same order they are described in the text.

In the first paragraph, the Tl and imd pathways are described as being specific for Gram+ versus Gram- bacteria (lines 378 ff), although most researchers these days agree they depend on the type of peptidoglycan (DAB vs Lys-type).

Finally, in the discussion there are parts that fit better into the introduction (for example lines 372-380 and 401-406.

In summary, although a substantial set of data are presented, the authors should be careful to not overstate the direct connection between hyperproteinemia and the immune defects and be more cautious about their conclusions. In addition the manuscript should be checked for language and grammar. 

Author Response

(The authors gave the same response as above.)

Reviewer 3 Report

    Authors previously developed an animal model for hyperproteinemia using the silkworm Bombyx mori. Using this animal model, they explore influences of hyperproteinemia on the innate immune system in the current study. They found that hyperproteinemia affects on the heamocyte population. They also showed that hyperproteinemia induced antibacterial activity against gram-negative bacteria of haemolymph probably through up-regulation of Toll and Imd signals. In contrast hyperproteinemia decreased melanization capability of haemolymph, which was restored by ecdysteroid injection. Although authors presented a series of interesting data, I am concerned for the publication about several points listed below.

Major points:

  1. Section 3.1.

The effect of hyperproteinemia on the proportion of haemocyte types was described, but the total number of each cell type is also important. This should be stated.

  1. Lines 162-165.

Authors described that active phagocytosis by granulocytes were clearly observed. I wonder whether Figure 1D is enough to show not only the envelopment of fat body cells but also phagocytosis. I also wonder whether these results suggest complicated changes of immune function of hemolymph as authors mentioned. The heamocytes might carry out just their natural roles for immune function.

  1. Section 3.2.

This section describes about transcriptomic data. But these transcriptomic analyses are not described in Materials and Methods. In Fig. 2A, there are two axes labeled "Percent of genes" and "Number of genes". It is necessary to explain how to read this data. In Fig. 2D, there are no mention about AM96h_1/2/3 and CK96h_1/2/3.

  1. Lines 192-196.

It is difficult to understand the discussion described in this paragraph. It is not clear which data the discussion is based on.

  1. Lines 230-238.

It is interesting that hyperproteinemia has different effect on gram-negative and gram-positive bacteria, even though the AMPs effective to gram-positive bacteria were also up-regulated. The possible reason of this difference should be addressed.

  1. Line 304.

What does "immune consumption" mean? It should be clearly stated.

  1. Lines 321-323.

Authors argues that melanization of heamolymph weakened by hyperproteinemia was rescued by 20E injection. However, it is known that 20E induces dopa decarboxylase (DDC) that is involved in the melanization. Therefore, weakened melanization by hyperproteinemia and strengthened melanization by 20E might be independent. The relationship amaong hyperproteinemia, 20E, and melanization is not clear.

  1. Lines 324-342.

In this paragraph, the topics about serine content and Spn27A started incoherently without connection to the preceding description. It is difficult to understand why they were analyzed, although these factors are explained in the Discussion section.

  1. Lines 423 and 430.

The evidences to state "feedback regulation" and "reduced sensitivity" are not clear to me. This paragraph (and the related paragraph mentioned above, lines 324-342) should be discussed more clearly.

  1. Fig. 7.

The figure shows that high PPC inhibits a signal molecule, which induces PPAE and PPOs in haemocyte. This resulted in the weakened melanization of haemolymph. However, authors showed that high PPC decreased the number of plasmacytes that produce phenoloxidase. Therefore, it is also possible that what high PPC affected was just the number of plasmacytes and it did not affect the signaling pathway for the phenoloxidase production in heamocytes.

Minor points:

  1. Line 70.

toll-like receptor should be Toll-like receptor.

  1. Line 85.

The citations (36-38) seem to be incorrect.

  1. Line 132.
  2. coli and S. aureus should not be abbreviated. This is the first appearance of these species.

  1. Line 195.

Table S1 should be Table S2.

  1. Fig. 4A.

It is necessary to mention clearly in the legend which samples were used. Is this the heamolymph from the animals in 192 h after modeling?

Author Response

(The authors gave the same response as above.)

Round 2

Reviewer 1 Report

To Authors,

Thank you for responding to my suggestions. You responded to my suggestions correctly. If you OK, please change your expression from “removed” to “collected” (line 164).

I hope this revised manuscript meets to criteria for publication to Biology.

Author Response

Thank you for your affirmation. We have tried our best to improve the manuscript and have made a lot of changes which we hope meet with approval.

Reviewer 2 Report

Major points: Line 226: I still have doubts about actin being a phagocytosis marker, is there a reference for that?

Line 387: why serine, it seems to me the authors talk mostly about serine proteases and serine-protease inhibitors, which are named after the serine in the active site and not about the amino-acid serine as a metabolite. Alternatively, and it seems this is what the authors want to imply, serine levels may be affected during immune reactions but then there should be added a reference for that, also in line 474.

Figure 4 why was the hemolymph heat treated (reference). If there is a reason for the  treatment, that should be explained (reference)

Minor points: Line 11: this is still not a complete sentence, delete “which”

Line 21:  delete “that”

Line 27: remain (Singular)

Line 87: granulocytes (plural)

Line 120: was (singular)

Line 124 delete “the”

Line 130: replace “which” with “when”

Line 140: was does “mixed” mean (each replicate included)?

Line 164 “was” (singular)

Line 172: my suggestion “melanization measurement”

Line 221: were (plural)

Line 315ff: this sentence still lacks a verb

Line 350: in “hemolymph immunity” or “in humoral immunity”

Line 418: my suggestion: and weakened melanization. (one does not have to mention immunity, please check also in other places “melanization immunity” is simply not used).

Author Response

(The authors gave the same response as above.)

Reviewer 3 Report

The revised manuscript responds well to the comments. I still have some concerns.

  1. Figure 1G.

Expression data for actin gene was added. Because B. morihas multiple genes for actin, it should be stated which actingene was analyzed. The primer sequences for the actin gene should be shown in Table S1. A revised version of supplemental materials were not attached in the reviewing website.

  1. Line 159.

Table S2 must be Table S1.

  1. Name of species

Names of species should be written in italics. Some of them are written in roman (e.g. Bombyx mori in line 79, Drosophila in line 92, Morus in line 116).

  1. Proofreading.

The manuscript should be proofread carefully. There are multiple inconsistency and misspellings (e.g. gram-positive and Gram-positive, and melanination in line 475). The current version with the revised sentences should be proofread/edited again by an English proofreading service.

Author Response

(The authors gave the same response as above.)
